



# Particle deliquescence in a turbulent humidity field

Dennis Niedermeier[1,*], Rasmus Hoffmann[1,*], Silvio Schmalfuss[1], Wiebke Frey[1,2], Fabian Senf[1], Olaf Hellmuth[1], Mira Pöhlker[1], and Frank Stratmann[1]

[1]Leibniz Institute for Tropospheric Research, Leipzig, Germany
[2]Wageningen University and Research, Wageningen, Netherlands
[*]These authors contributed equally to this work.

**Correspondence:** Dennis Niedermeier (dennis.niedermeier@tropos.de)

**Abstract.** The atmosphere contains aerosol particles, some of which are hygroscopic in nature. These particles have direct and indirect effects on weather and climate. Furthermore, turbulence causes fluctuations in temperature, water vapor content, and relative humidity (RH). Turbulent humidity fluctuations may influence, among others, the phase state of specific hygroscopic particles. One process of particular interest in that context is particle deliquescence which is the phase transition of solid

particles to solution droplets. It occurs at a certain RH, the so-called deliquescence relative humidity (DRH), which in turn depends on e.g., the particle substance. This study investigates the deliquescence behavior of sodium chloride particles in a turbulent humidity field, in particular addressing the questions whether and how turbulent relative humidity fluctuations affect the number / number fraction of deliquesced particles. The turbulent moist air wind tunnel LACIS-T (Turbulent Leipzig Aerosol Cloud Interaction Simulator) is used for this study. The results show that the number of deliquesced particles is influenced by

turbulent RH fluctuations. On the one hand, particle deliquescence can be observed although the mean RH is smaller than DRH. On the other hand, there are cases for which non-deliquesced particles are present even though the mean RH is larger than DRH. In general, the number fraction of deliquesced particles depends on a combination of mean relative humidity, strength of humidity fluctuations, and residence time of the particles in the turbulent humidity field. The study concludes that relying solely on the mean relative humidity is inadequate for determining the phase state of deliquescent particle species in

the atmosphere. It is necessary to additionally consider both the humidity fluctuations and the particle history.

## 1 Introduction

A large portion of the atmospheric aerosol consists of hygroscopic particles, which feature different sizes and phase states, and may have immense impacts on Earth's weather and climate. The particle size affects, for example, particle radiative properties as well as the particles' potential to function as cloud condensation nuclei (CCN). The phase state also influences particle radia-

tive properties, as hygroscopically grown particles have different angular scattering properties and refractive indices compared to their dry counterparts (Titos et al., 2016). The liquid particle fraction scatters more light than the respective solid fraction (e.g., Toon et al., 1976; Sloane, 1984). Furthermore, the phase state influences gas-particle partitioning, heterogeneous and multi-phase chemistry processes as, for example, the presence of water allows for reactions with atmospheric pollutants (e.g., Finlayson-Pitts and Hemminger, 2000; Bahadur and Russell, 2008; Liu et al., 2019).



The phase state depends on the particle properties, i.e., chemical composition and size, as well as the atmospheric conditions, such as temperature and relative humidity (RH). One process of particular interest in that context is particle deliquescence, for which a solid, water-soluble particle turns into an aqueous solution droplet, thereby increasing its size significantly (Khvorostyanov and Curry, 2014). This phase transition occurs at a certain RH, called deliquescence RH (DRH), and is specific for each deliquescent particle substance. The DRH also depends on temperature (Seinfeld and Pandis, 2006; Khvorostyanov

and Curry, 2014) and particle size (Bahadur and Russell, 2008). Beyond the DRH, a further increase of the RH leads to the growth of the formed solution droplet. However, if the RH decreases below DRH, the material dissolved in the solution droplet will not re-crystallize. The solution will become supersaturated and will remain in a metastable state until it reaches another specific RH, called the efflorescence RH (ERH, with ERH < DRH) at which re-crystallization, i.e., efflorescence occurs. This behavior leads to a hysteresis curve which implies that the phase state of a soluble, deliquescent particle, which is at a RH

between ERH and DRH, depends on its history (Seinfeld and Pandis, 2006; Khvorostyanov and Curry, 2014; Titos et al., 2016).

     A wide range of theoretical and experimental studies have been performed in the past and the obtained results significantly increased both the fundamental and quantitative understanding of aerosol particle deliquescence (e.g., Seinfeld and Pandis, 2006; Shchekin et al., 2008, 2013; Hellmuth et al., 2013; Khvorostyanov and Curry, 2014; Hellmuth and Shchekin , 2015; Tang

et al., 2019; Peng et al., 2022). This holds even for particles with complex chemical composition. Most of the experimental investigations have focused on the process itself using various techniques (Tang et al., 2019). Few experiments were performed under laminar flow conditions (e.g., Wex et al., 2007). However, the atmosphere is turbulent and turbulent mixing leads to strong fluctuations in temperature, water vapor concentrations, and consequently RH (Siebert et al., 2006; Bodenschatz et al., 2010), which could affect the phase state of deliquescent particles. To our knowledge, the behavior of deliquescent particles in

turbulent humidity fields has not yet been investigated in detail. The questions are whether and how turbulent RH fluctuations affect the number of deliquesced particles, and whether or not the number of deliquesced particles is time dependent due to the hysteresis effect.

     With the turbulent moist-air wind tunnel LACIS-T (Turbulent Leipzig Aerosol Cloud Interaction Simulator, Niedermeier et al. (2020)), we wanted to address these fundamental questions. We performed various experiments for different mean RHs,

RH fluctuation intensities as well as residence times, and determined the number fraction of deliquesced particles. We used size-selected, monodisperse NaCl particles in this study. NaCl was chosen as it is an atmospherically relevant substance being, e.g., the main component of sea salt particles (Niedermeier et al., 2008). Furthermore, the deliquescence properties of NaCl are well characterized: The DRH at 15°C, the temperature used in our experiments, is about 75.5% (Seinfeld and Pandis, 2006), the temperature dependence of the DRH is weak for NaCl (Seinfeld and Pandis, 2006; Khvorostyanov and Curry, 2014), and

for a given temperature, DRH is constant for particle diameters larger than 100 nm (Bahadur and Russell, 2008).

     The experimental investigations at LACIS-T were accompanied by fluid dynamics simulations performed with OpenFOAM. These large-eddy simulations (LES) assist in setting up the experiments and aid in interpreting the observations. For example, the simulations provide the strength of RH fluctuations that cannot be obtained experimentally.





## 2 Experimental setup

This section describes the generation, size selection as well as pre-conditioning of the NaCl particles, and introduces the basic functionality of LACIS-T, the investigated parameter space, as well as the applied instrumentation.

### 2.1 Particle generation, size selection and pre-conditioning

The NaCl particles were generated by means of an atomizer (TSI 3075, TSI Inc., St. Paul, Minnesota, USA) atomizing an aqueous solution of 1 g NaCl per liter of double de-ionized water. The resulting aerosol was dried in a diffusion dryer to a RH
lower than 20%, i.e., a RH well below the ERH of NaCl particles, which is between 43% and 45% for the investigated particle size range (Tang et al., 1977; Cziczo et al., 1997; Gao et al., 2007).

Downstream of the dryer, the particles are charged by means of a neutralizer. A Differential Mobility Analyzer (DMA, Knutson and Whitby (1975), type "Vienna medium") is used to select a narrow particle size fraction. Inside the DMA, a further drying occurred as the RH of the DMA sheath air was always $\leq 5\%$. For the experiments, we selected a mobility diameter of
$Dp_{\mathrm{mob}} = 400\,\mathrm{nm}$ in order to be able to detect dry solid particles optically by means of a Promo 2000 with welas 2300 aerosol spectrometer (Palas GmbH, Karlsruhe, Germany) inside LACIS-T. Note that a particle shape factor has to be considered when converting the mobility diameter to a mass equivalent diameter, because of the non-spherical shape of solid NaCl particles. This factor is 1.08 for NaCl particles according to Kelly and McMurry (e.g., 1992). Earlier measurements with the laminar flow-tube LACIS (Wex et al., 2005; Niedermeier et al., 2008) confirm that this shape factor is valid for NaCl particles in the
selected size range. In consequence, $Dp_{\mathrm{mob}} = 400\,\mathrm{nm}$ corresponds to a mass equivalent diameter of $Dp_{\mathrm{me}} = 370\,\mathrm{nm}$.

The number concentration of the selected particles was determined utilizing a Condensational Particle Counter (CPC, TSI 3010, TSI Inc., St. Paul, Minnesota, USA), and was kept at about $1000\,\mathrm{cm}^{-3}$ by means of a dilution system upstream of the DMA. The dilution system consists of a by-pass, a filter and two valves for adjustment of the flows. All flows, i.e., aerosol flow, DMA sheath, and excess air-flow as well as CPC sample flow were controlled by mass flow controllers (Brooks Instrument
GmbH, Dresden, Germany) and checked with a bubble flow meter (Gilian® Gilibrator™2, Sensidyne Inc., Clearwater, Florida, USA) on a daily basis.

For the experiments, three different types of pre-conditioning were applied concerning the NaCl particles which were fed into LACIS-T:

Case (i): The particles were left dry (i.e., RH $\leq 5\%$) so that solid NaCl particles entered the measurement section of
LACIS-T. This setup was only used to determine the optical size distribution of solid NaCl particles (see Sect. 4.2), as dry particle injection into LACIS-T reduces the mean RH in the mixing zone of the measurement section significantly (see Appendix A).

Case (ii): The particles were pre-humidified to a dew-point temperature of $T_{\mathrm{d}} = 9.6°\mathrm{C}$ ($\pm 0.1°\mathrm{C}$) by mixing the dry aerosol with pre-humdified particle-free air by means of a saturator (Perma Pure MH-110-12S-4, Perma Pure LLC,
Toms River, New Jersey, USA). This led to a RH of 70.2% ($\pm 0.3\%$) at 15°C, i.e., the NaCl particles were still non-deliquesced before entering LACIS-T. With RH at the LACIS-T inlet being closer to DRH, the mean RH inside the





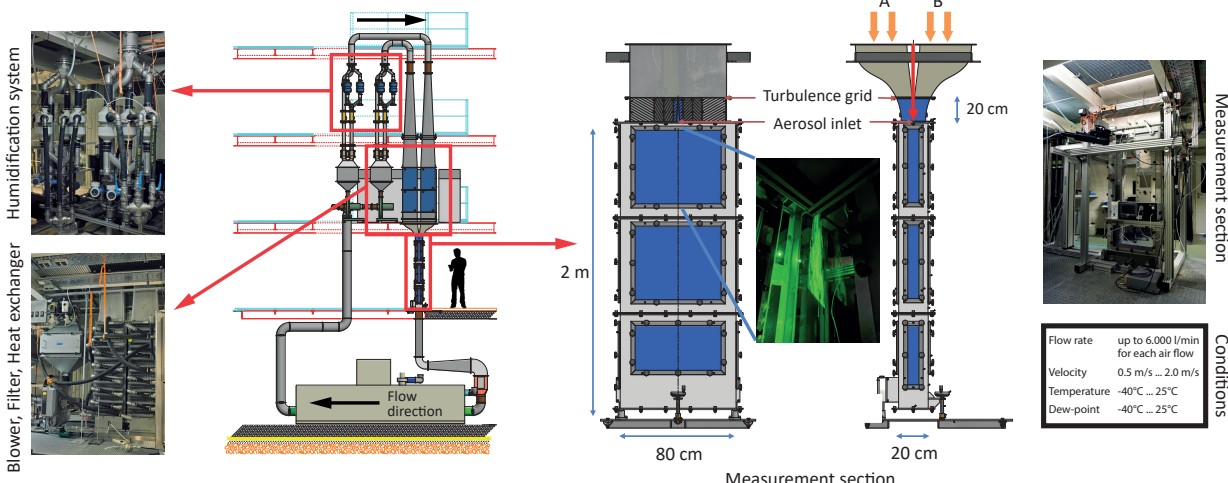

**Figure 1.** Schematic of LACIS-T including main components and possible conditions. Schematic copyrights: Ingenieurbüro Mathias Lippold, VDI; TROPOS. The figure is adapted from Niedermeier et al. (2020) and modified.

measurement section of LACIS-T is not reduced significantly (see LES results in Sect. 4.1). This setup was used for the LACIS-T experiments for different sets of mean RH as well as strength of RH fluctuations.

Case (iii): The particles were pre-humidified at RH = 100% ($T_d = 12.5°C$ ($\pm 0.1°C$)) by means of a saturator (Perma Pure MH-110-12S-4, Perma Pure LLC, Toms River, New Jersey, USA) so that the NaCl particles deliquesce before entering LACIS-T (RH = 85.0% at 15°C). This setup was used to determine the optical size distribution of fully deliquesced NaCl particles inside LACIS-T for the different sets of conditions.

## 2.2 LACIS-T

LACIS-T is a unique turbulent moist-air wind tunnel which has been established to study cloud physical processes, in general, and cloud microphysics – turbulence interactions, in particular. LACIS-T can be operated under a wide range of well-characterized and reproducible initial and boundary conditions resembling atmospheric warm, mixed-phase and cold clouds. The design, functionality and capabilities of the setup are described in detail in Niedermeier et al. (2020) and only a brief description will be given here.

LACIS-T is a closed-loop wind tunnel of Göttingen type (Fig. 1). The unique characteristic of LACIS-T is that it features two parallel flow branches, in which two particle-free air-flows 'A' and 'B' can be separately conditioned and controlled with respect to volume flow rate (through the radial blowers and valves, up to 6000 l/min each), water vapor content in terms of a dew-point temperature (through the humidification system, dew-point temperature range $-40°C < T_d < 25°C$, with an accuracy of 0.1 K), and temperature (through the heat exchangers, temperature range $-40°C < T < 25°C$, with an accuracy of



0.05 K). These two air-flows pass passive square-mesh grids through which defined turbulence is induced. They then enter the
measurement section, at the entrance of which the particle-free air-flows are combined and turbulently mixed. The measurement
section is of cuboid shape with the dimensions of 200 cm x 80 cm x 20 cm. The aerosol particles are introduced into the mixing
zone of the two turbulent particle-free air-flows. This mixing zone provides an ideal environment for studying the influence of
the turbulent fluctuations on aerosol and cloud microphysical processes. Downstream of the measurement section, the flow is
directed towards an adsorption dehumidifying system where it is dried and heated. Afterwards, the flow is split up into the two
branches, cleaned by particle filters, and the whole cycle starts again.

In this study, the flow rate was set to 4300 l/min for each particle-free air-flow leading to a mean flow velocity of 1.35 m/s
in the measurement section. Turbulence conditions inside the measurement section are determined by means of Hot Wire
anemometry (Dantec Dynamics Inc., Skovlunde, Denmark) and LES and are very similar to those described in Niedermeier
et al. (2020). For example, the eddy turnover time $\tau_{\mathrm{mix}}$ which is a measure for the turbulent mixing time scale, is between
$\tau_{\mathrm{mix}} = 0.1$ - $0.7$ s. It increases due to the decaying turbulence inside the measurement section and is given as $\tau_{\mathrm{mix}} = (l_{\mathrm{T}}^2/\varepsilon)^{(1/3)}$
(e.g., Baker et al., 1984; Lehmann et al., 2009). The quantity $\varepsilon$ represents the energy dissipation rate, while $l_{\mathrm{T}}$ denotes the Taylor
microscale. In the context of atmospheric conditions, the Taylor microscale is frequently considered to be the appropriate
mixing length scale (Lehmann et al., 2009).

We use isothermal conditions in this study, i.e., the temperature in both particle-free air-flows was identical, set to $T_{\mathrm{A}} = T_{\mathrm{B}}$
= 15°C and monitored via PT100 temperature sensors (1/10 class B, DIN EN 60751, accuracy of $\pm(0.0300°C +0.0005 \times T)$).
The dew-point temperature was set individually in each particle-free airflow and monitored by means of two dew-point mirrors
(DPM, model 973 by MBW Calibration AG, accuracy of $\leq \pm0.1$ K; reproducibility of $\leq \pm0.05$ K), sampling air in each flow
branch between heat exchanger and turbulence grid. Due to the individual settings of the dew-point temperatures, and thus the
individual RH values in each flow branch, various RH mixing conditions could be established. That means different mean RH
values, and different strengths of RH fluctuations, could be set in the mixing region (see Fig. 2a as an example for a specific
set of RH conditions in branch A and B). In general, the larger the difference between $RH_{\mathrm{A}}$ and $RH_{\mathrm{B}}$, the larger the strength of
the RH fluctuations. However, no direct measurement of dew point temperature fluctuations and consequently RH fluctuations
is available so far. The strengths of the fluctuations are determined by means of the LES which is described in Sect. 4.1.

Finally, particle size distributions are determined optically via a Promo 2000 with welas 2300 aerosol spectrometer (Palas
GmbH, Karlsruhe, Germany). The welas 2300 sensor is placed inside the measurement section having a 15 cm long stainless-
steel tube (5 mm inner diameter) on top of its own inlet tube. This additional long tube ensures particle extraction and detection
from the flow field, which is only very weakly influenced by the body of welas 2300, which disturbs the flow field in its closer
vicinity. Please note that the distance of the tip of this stainless-steel tube relative to the aerosol inlet of LACIS-T, called '$z$', is
used as a reference in the later data description.

In the first set of experiments, welas 2300 is placed at a fixed position inside the measurement section leading to a fixed
residence time of the particles until the detection occurs. The inlet tube is placed at $z = 30$ cm below the aerosol inlet of
LACIS-T. In the second set of experiments, the position of the tube together with the welas 2300 is varied leading to different
mean residence times of the particles.





## 3 Numerical setup

Large-eddy simulations directly compute the large-scale motions of the flow fields from the Navier-Stokes equations, and parameterize the unresolved small-scale/sub-grid scale motions. We use the dynamic k-equation LES model in OpenFOAM based on Chai and Mahesh (2012), who developed a new transport equation for the subgrid-scale kinetic energy, which has proven to be a good model for decaying turbulence and the transport of thermodynamic quantities. A hexaeder dominated mesh is applied, refined near turbulence grid and walls, with $7.6 \cdot 10^6$ cells in total. We use an Euler-Lagrange approach for the simulations, i.e., temperature and water vapor mixing ratio are calculated at the grid points while each individual particle is tracked along its trajectory through the thermodynamic field. Details about the general numerical setup, especially about how to set the initial and boundary conditions, are given in Niedermeier et al. (2020).

Overall, the main goals of these simulations were a) the determination of the RHs the particles experience along their way through the measurement section for the different experimental conditions, and b) the determination of the deliquesced particle fraction for comparison with the gained experimental data. The RHs, and RH fluctuations the particles experience, which cannot be detected experimentally, were quantified in terms of mean standard deviation of simulation-derived RH probability density functions (pdfs). Thereto, the RH pdfs are calculated for specific locations along the mixing region inside LACIS-T's measurement section. Furthermore, the numbers of deliquesced and non-deliquesced particles were derived from the simulations and compared with the experimentally obtained ones. Concerning the latter, a dynamical growth law is implemented for the Lagrangian particle simulations accounting for diffusive vapor transport and deposition to the particle surface, and latent heat release (Wilck, 1999) as well as deliquescence and efflorescence. However, the feedback effects on heat transport and mass transfer in the gas phase are ignored as the droplet concentrations (mass and volume ratio) considered in this study are too low to influence the continuous phase. As we are modeling the particle hygroscopic growth dynamically, the model requires information concerning the deliquescence time scale for the considered NaCl particles. We assumed deliquescence to occur on the time scale $\tau_{del}$ of $10^{-4}$ s (the numerical time step is also $10^{-4}$ s) once DRH = 75.5% is reached (Note that this time scale is 3 orders of magnitude smaller than the mixing time scale $\tau_{mix}$ which is between 0.2 and 0.7 s). This assumption is based on molecular dynamics simulations performed by Bahadur and Russell (2008) who give the time scale $\tau_{del}$ required for complete deliquescence of an infinitely extended "planar" NaCl slab to be $\tau_{del}$ = 9.96 $10^{-5}$ s. The extent to which this assumption is reasonable will be the subject of discussion in Appendix B.

For particle efflorescence, we also assume that the re-crystallization occurs on the time scale of $10^{-4}$ s once ERH = 45.0% is reached. Tang and Munkelwitz (1984) show that the re-crystallization of a single micrometer-sized NaCl particle can occur very quickly, however, a clear time scale cannot be determined from Fig. 3 in their paper. Other studies like Ma et al. (2019) show that the time scale for particle efflorescence, which is a nucleation process, depends on the ambient RH and is inversely proportional to the nucleation rate. Various studies show that the ERH of NaCl particles is between 43% and 45% for the investigated particle size range (Tang et al., 1977; Cziczo et al., 1997; Gao et al., 2007). As it turned out, the minimum RH reached in our experiments is well above 45%. From this point of view, our assumption about the time scale of particle efflorescence will supposably not affect the simulation results as efflorescence is unlikely to occur in our investigations.





## 4 Results

This section presents the results of the numerical and experimental investigations. First of all, the LES provide the RH ex-
perienced by the particles. This is then applied to determine the strength of the RH fluctuations which are later used for the
interpretation of the experimental results. The experiments themselves are performed for different mean RH, strengths of RH
fluctuations, as well as for fixed and variable particle residence times. Number fractions of deliquesced particles are determined
and discussed.

### 4.1 Turbulent RH field obtained via LES

At first, the simulations are employed to determine the RH experienced by the particles, as well as to ascertain the impact of the
particles themselves on the RH field within the mixing region inside the measurement section. This is presented exemplarily for
the setting of $RH_A = 60\%$ and $RH_B = 85\%$ leading to a mean RH of 72.5% along the center line. About 78.000 monodisperse
NaCl particles with a mass equivalent diameter of $Dp_{me} = 370$ nm are tracked inside the measurement section. The NaCl
particles are pre-conditioned according to case (ii), i.e., the aerosol is injected with $RH = 70.2\%$ (as introduced in sect. 2.1). In
Fig. 2a, a snapshot of the instantaneous RH field in the symmetry plane is shown including the respective particles deliquescing
and growing along the inverse vertical axis. The dashed line represents the center line, where the strongest mixing occurs, as
well as where the particles are detected further downstream. As the particles shown move through the channel, statistics are
done at distinct horizontal planes over time, i.e., data from particles that pass a given horizontal plane is stored and later on
analyzed.

As mentioned, the RH of the introduced aerosol is slightly lower than the mean RH along the center line of the measurement
section. This influences the RH field along the center line, which is illustrated in Fig. 2b in terms of median RH ($RH_{med}$),
mean RH ($RH_{mean}$) and standard deviation of the RH ($\sigma_{RH}$). First of all, it can be seen that generally $RH_{med}$ and $RH_{mean}$ fall
together. Secondly, $RH_{mean}$ is observed to be slightly lower close to the location of aerosol injection. However, it increases on
the very first centimeters and then reaches a constant value of 72.5% at a height of approximately $z = 15$ cm. Additionally,
$\sigma_{RH}$ demonstrates an increase within the initial 15 cm, subsequently reaching a constant value of 4.9%. Consequently, the
data obtained for $z \geq 15$ cm will be employed for the subsequent data interpretation. As an example, the RH experienced by
particles near the center line when passing the horizontal plane at $z = 30$ cm is shown as normalized frequencies in Fig. 2c
and the obtained RH distribution has a Gaussian shape.

### 4.2 Obtained particle size distributions and determined deliquesced particle fraction

In order to be able to determine the fraction of deliquesced particles, first the size distributions of both, dry and deliquesced
particles, as well as those of a mixture of both need to be considered. Therefore, in a first set of experiments, different RH
conditions are adjusted in order to obtain and interpret the corresponding size distributions. The corresponding $RH_{mean}$ values
at the center line are calculated from the RH values of the individual particle-free airflows. In general, it turns out from repeated
measurements of temperature and dew-point temperature inside the tunnel that there is an absolute uncertainty in $RH_{mean}$ of





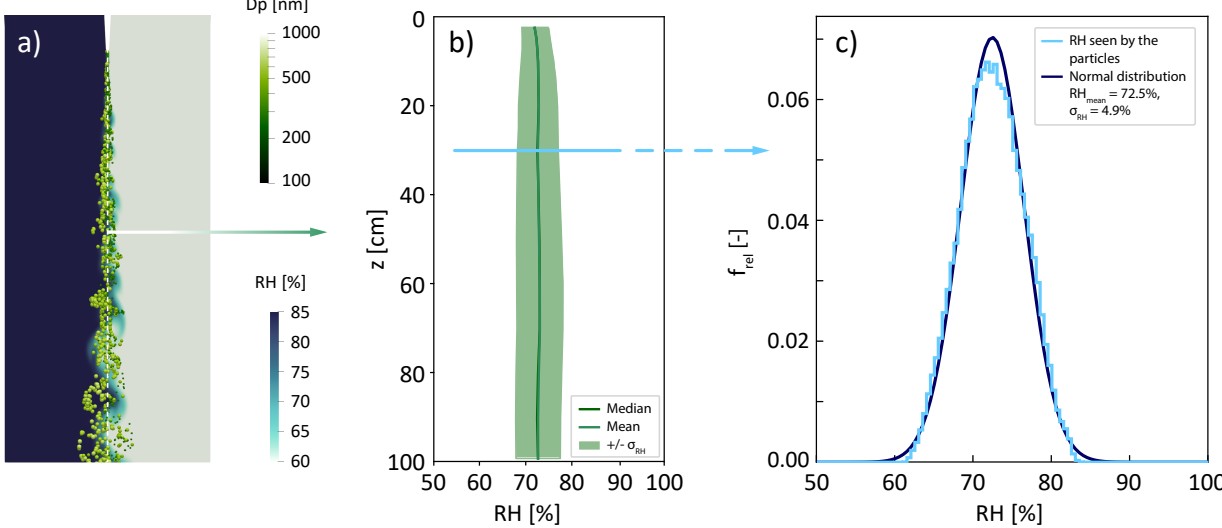

**Figure 2.** a) A snapshot of the particle simulation with the fluctuating RH field in the background (flow direction from top to bottom). It depicts the scenario of $RH_A = 60\%$ (right side) and $RH_B = 85\%$ (left side) resulting in a mean RH of 72.5% along the center line (dashed white line). Note that only 10% of the particles are depicted. The particles are colored and sized (not to scale) according to their diameter. b) The obtained temporal median RH, mean RH, and RH fluctuations in terms of $\pm\sigma_{RH}$ plotted along the center line as a function of the distance $z$ to the aerosol inlet of the measurement section. Note that the pre-humidified aerosol is injected (case(ii): RH = 70.2% at 15°C). c) The normalized frequency of the RH experienced by the particles. The distribution has a Gaussian shape as evidenced by the normal distribution with $RH_{mean} = 72.5\%$ and $\sigma_{RH} = 4.9\%$.

up to 0.6%. The width of the RH distribution which is given in terms of $\sigma_{RH}$ is obtained from the LES. For this first set of experiments, the welas 2300 position was fixed with its inlet being placed at $z = 30$ cm leading to a mean residence time of the particles before detection of approx. 0.22 s as the mean velocity is 1.35 m/s. Figure 3 shows the results of four different experiments for the conditions given in Table 1.

No distinct difference in the obtained particle size distributions is observed for experiment (1) and (2). During experiment
(2), the particles most likely start to take up water molecules leading to a thin liquid shell surrounding the particles, however, particles are still non-deliquesced under these humid conditions. This is in agreement to observations obtained by Krueger et al. (2003) and Wise et al. (2008) for single NaCl particles.

Experiment (3) shows a clear increase in particle size compared to (1) and (2). In this case, the NaCl particles are fully deliquesced. The mean particle diameter determined optically agrees with the diameter calculated with the Köhler equation
(Niedermeier et al., 2008) for this $RH_{mean}$.



**Table 1.** NaCl particle and LACIS-T conditions for four different experiments. Column 1 gives the Experiment number. Column 2 shows the condition of the NaCl particles before being inserted into LACIS-T. Columns 3 and 4 show the corresponding RH conditions inside LACIS-T in terms of $RH_{mean}$ and $\sigma_{RH}$.

| Experiment # | NaCl particle conditions* | LACIS-T conditions | |
|---|---|---|---|
| | | $RH_{mean}$ [%] | $\sigma_{RH}$ [%] |
| (1) | case (i) - solid, dry | 20.0 | 0 |
| (2) | case (ii) - solid, pre-humidified | 72.5 | 0 |
| (3) | case (iii) - fully deliquesced | 72.5 | 4.9 |
| (4) | case (ii) - solid, pre-humidified | 72.5 | 4.9 |

* see Sect. 2.1 for details

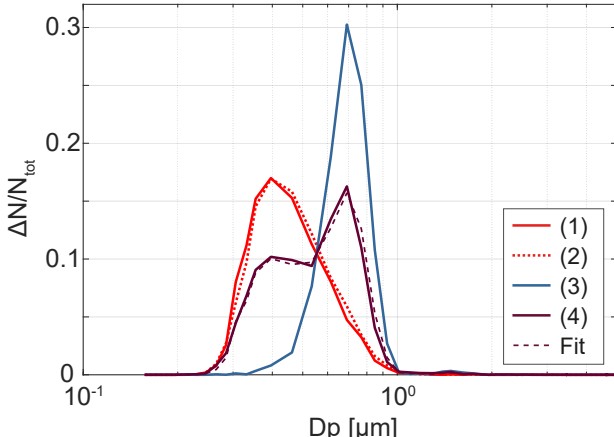

**Figure 3.** Normalized particle number size distributions determined for the four different experiments being summarized in Table 1. Experiments (1), (2), and (3) are given by the solid red, dotted red, and solid blue lines, respectively. Note that experiment (4) leads to a bimodal distribution @ $RH_{mean} = 72.5\%$, $\sigma_{RH} = 4.9\%$ featuring both solid and deliquesced NaCl particles (brown solid curve). The dashed brown curve represents a bimodal fit to curve of experiment (4) in order to determine the solid and deliquesced particle fractions.

Finally, experiment (4) results in two modes. The first mode fits with the one for solid NaCl particles, while the second mode fits with the one obtained for the deliquesced particles. That means, we observe both solid and deliquesced particles at the same time. In other words, the humidity fluctuations lead to particle deliquescence although $RH_{mean}$ is smaller than DRH which is about 75.5% at 15°C (Seinfeld and Pandis, 2006). By fitting both modes with two log-normal distributions (see Fig. 3) the solid and deliquesced particles fractions can be determined which in the example are 0.57 and 0.43, respectively.





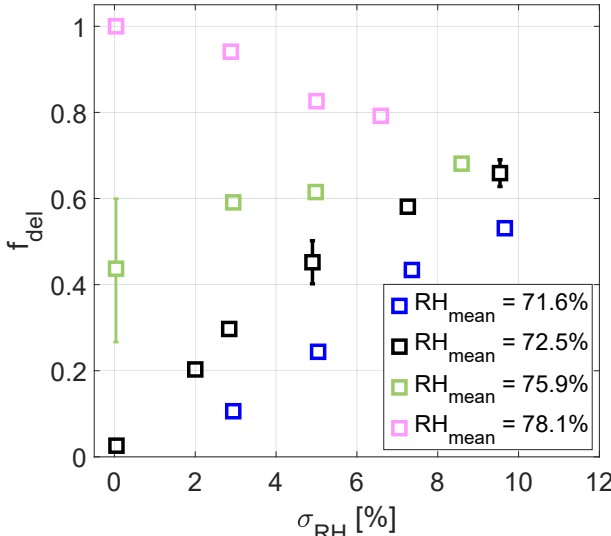

**Figure 4.** Deliquesced particle fraction $f_{\mathrm{del}}$ as a function of the standard deviation of the RH distribution $\sigma_{\mathrm{RH}}$. The width of the symbols represents the range of uncertainty of $\sigma_{\mathrm{RH}}$ based on the LES results. The shown error bars originate from repeated measurements (three to five times) giving the minimal and maximal obtained $f_{\mathrm{del}}$, respectively.

To systematically quantify this effect, such experiments have been performed for various thermodynamic and flow conditions, as well as particle residence time (i.e., welas 2300 sensor positions).

### 4.3 Dependence of deliquesced particle fraction on RH field and residence time

First of all, the influence of RH fluctuations on the deliquesced particle fraction was determined for a fixed welas 2300 position,
i.e. a fixed mean residence time. Thereto, the welas 2300 inlet was placed again at $z = 30$ cm. $\mathrm{RH_{mean}}$ was varied between 71.6% and 78.1%, i.e., from below to above DRH of the investigated NaCl particles, and $\sigma_{\mathrm{RH}}$ was set between 0% and 9.5%. The obtained deliquesced particles fractions $f_{\mathrm{del}}$ are depicted in Fig. 4 as a function of $\sigma_{\mathrm{RH}}$ for four different $\mathrm{RH_{mean}}$ values.

Generally, it becomes obvious that the humidity fluctuations have a strong influence on the fraction of deliquesced particles. Particle deliquescence can be observed although $\mathrm{RH_{mean}} < \mathrm{DRH}$. Furthermore, we detect an increase of the deliquesced
particle fraction with increasing $\sigma_{\mathrm{RH}}$ as long as $\mathrm{RH_{mean}} < \mathrm{DRH}$. In this case, an increasing $\sigma_{\mathrm{RH}}$ increases the probability that solid NaCl particles are in a RH field with RH-values higher than DRH. The slope of $f_{\mathrm{del}}$ flattens for $\mathrm{RH_{mean}}$ being close to DRH and becomes negative for $\mathrm{RH_{mean}} > \mathrm{DRH}$ as now the increasing $\sigma_{\mathrm{RH}}$ increases the probability that solid NaCl particles experience a RH field with RH-values lower than DRH, and therefore do not deliquesce. In other words, in this case not all particles deliquesce although $\mathrm{RH_{mean}} > \mathrm{DRH}$.





Secondly, the influence of RH fluctuations on the deliquesced particle fraction was determined as a function of mean particle residence time $t_{\mathrm{res}}$ in a fluctuating RH field. This was achieved by changing the position of the welas 2300. The purpose of this type of experiment is as follows. Particles once deliquesced will not recrystallize / effloresce at DRH. They would do so at the ERH, which is between 43% and 45% for NaCl particles of the investigated size range (Tang et al., 1977; Cziczo et al., 1997; Gao et al., 2007). However, in our experiments, even for the broadest RH distributions we do not reach this ERH. That

means if solid NaCl particles are inserted in fluctuating RH fields, the residence time in this field might influence the number and fraction of deliquesced particles.

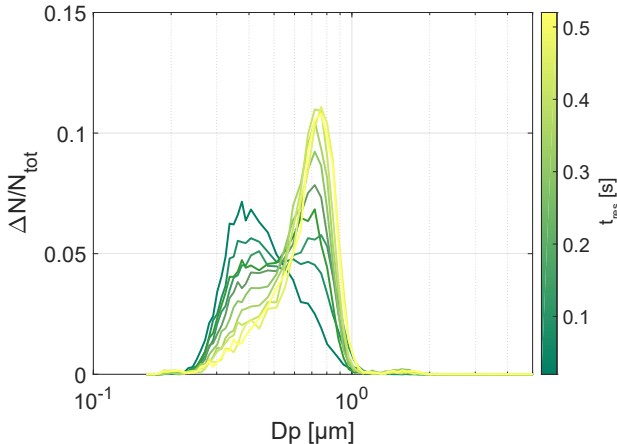

**Figure 5.** Normalized particle number size distributions determined experimentally for $\mathrm{RH}_{\mathrm{mean}} = 75.9\%$ and $\sigma_{\mathrm{RH}} = 4.9\%$, but for different residence times $t_{\mathrm{res}}$ inside the measurement section. The colors of the different number distributions correspond to the colormap on the right-hand side of the plot denoting $t_{\mathrm{res}}$.

Figure 5 shows a series of normalized particle size distributions for $\mathrm{RH}_{\mathrm{mean}} = 75.9\%$ and $\sigma_{\mathrm{RH}} = 4.9\%$, with different $t_{\mathrm{res}}$ inside the measurement section. It can be seen that the deliquesced particle mode of the distributions (right mode) increases with increasing $t_{\mathrm{res}}$ while the solid particle mode (left mode) decreases. In other words, more particles deliquesce with increasing

$t_{\mathrm{res}}$ due to the hysteresis effect.

Measurements as presented in Fig. 5 have been carried out for other sets of $\mathrm{RH}_{\mathrm{mean}}$ and $\sigma_{\mathrm{RH}}$ values. The respective results are summarized in Fig. 6. Here, the fraction of deliquesced particles $f_{\mathrm{del}}$ is depicted as a function of $t_{\mathrm{res}}$ (logarithmic scale) inside the fluctuating RH environment. In the left plot, $\mathrm{RH}_{\mathrm{mean}}$ is fixed and $\sigma_{\mathrm{RH}}$ is varied. In the right plot, it is vice versa. The key findings are that the fraction of deliquesced particles increases with increasing $t_{\mathrm{res}}$ for all investigated cases. For fixed

$\mathrm{RH}_{\mathrm{mean}}$, the slope of the three curves looks very similar pointing towards an approximately exponential relationship between $f_{\mathrm{del}}$ and $t_{\mathrm{res}}$, with the curves shifting upwards to higher values of $f_{\mathrm{del}}$ as $\sigma_{\mathrm{RH}}$ increases. When looking at the curves for fixed



$\sigma_{\mathrm{RH}}$ it can be observed that the slopes become flatter with increasing $\mathrm{RH}_{\mathrm{mean}}$, as a greater proportion of the particles is already deliquesced at the lowest measured residence times. Consequently, fewer solid particles are available for deliquescence.

In summary, it can be said that the combination of turbulent RH fluctuations and hysteresis in aerosol particle deliquescence
and efflorescence, has a significant impact on the fraction of deliquesced particles over time. The time required for all NaCl particles to deliquesce depends on both the mean RH and the strength of the fluctuations, or in other words, the proportion of deliquesced particles is dependent on $\mathrm{RH}_{\mathrm{mean}}$, $\sigma_{\mathrm{RH}}$, and residence time.

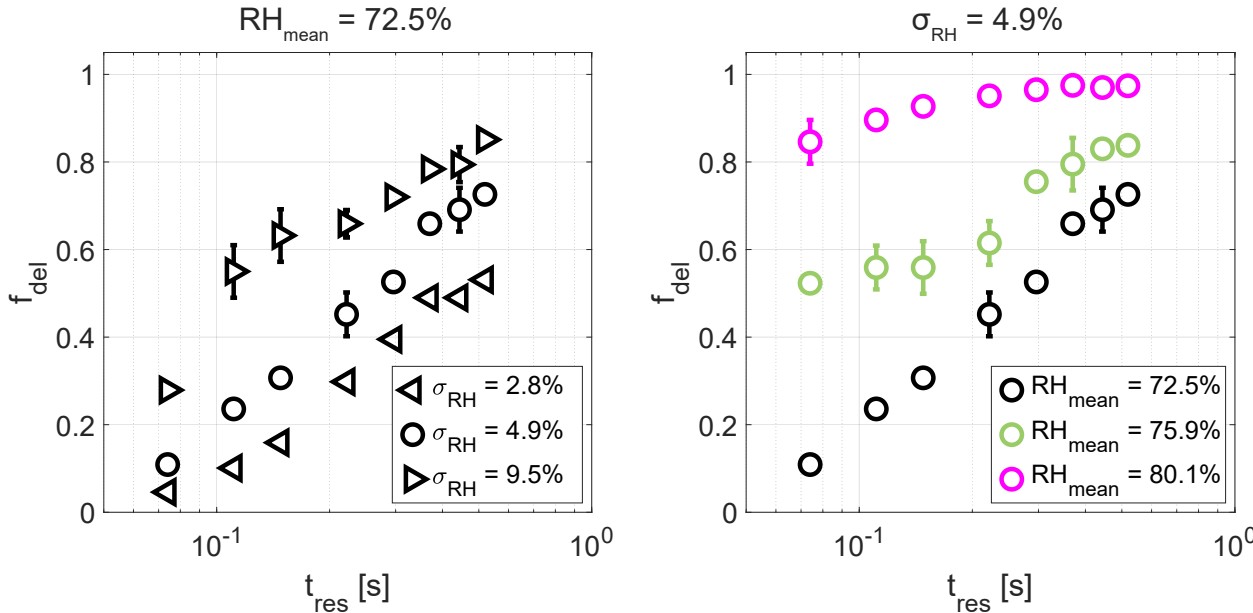

**Figure 6.** Deliquesced particle fraction $f_{\mathrm{del}}$ as a function of particle residence time $t_{\mathrm{res}}$. Left figure: $\mathrm{RH}_{\mathrm{mean}}$ was set to 72.5% and $\sigma_{\mathrm{RH}}$ was varied. Right figure: $\sigma_{\mathrm{RH}}$ was set to 4.9% and $\mathrm{RH}_{\mathrm{mean}}$ was varied.

Finally, we simulated the deliquescence of the NaCl particles in the turbulent RH field by running LES in OpenFOAM. In general, the time dependence of $f_{\mathrm{del}}$ can also be identified in the model results. However, the simulations appear to overes-
timate the observed deliquesced particle fractions, which might be caused by our assumption of a constant time scale for the deliquescence process itself. A detailed discussion about this discrepancy and its possible reasoning is given in Appendix B. Nevertheless it is important to point out that this discrepancy between measurement and simulation does not invalidate the experimental observations and central statements. Furthermore, it motivates us to investigate the suitability of different theories in the future (more details given in Appendix B).



## 5 Summary and conclusion

We investigated the deliquescence behavior of size-selected, monodisperse NaCl particles in a turbulent humidity field with LACIS-T. The mean RH, the strength of RH fluctuations and the residence time of the particles in the turbulent humidity field were varied. In general, we found that turbulence affects the number of deliquesced particles in a particle population and this number depends on the combination of all three of the aforementioned variables. Fluctuations in RH can lead to particle deliquescence, despite the mean RH being below the deliquescence RH. Conversely, particle deliquescence can be hindered even though the mean RH is above the deliquescence RH. However, a population of solid, non-deliquesced NaCl particles introduced into a fluctuating RH field, where the RH is always greater than the efflorescence RH, and the RH fluctuations exceed DRH will deliquesce completely due the combination of the turbulent RH fluctuations and the hysteresis effect. The time required to reach this fully deliquesced state is contingent upon the mean RH and the strength of the fluctuations.

We are able to simulate the general behavior of the NaCl particle deliquesence in the turbulent RH field by running LES in OpenFOAM. However, the simulations tend to overestimate the observed deliquesced particle fractions, which might be caused by our assumption of a constant time scale for the deliquescence process itself. In order to achieve a more accurate representation of the experimental observations, different theories for the derivation of the deliquesence time scale $\tau_{del}$ will be tested in the future. Following, for example, the argumentation of Khvorostyanov and Curry (2014), nucleation rates, and with that characteristic deliquescence time scales - which depend on RH among other things - will be determined and implemented into the LES model.

Ultimately, the observation of the onset of particle deliquescence below, and the presence of non-deliquesced particles above the DRH in a turbulent humid field is inline with the argumentation of Prigogine (1979), according to which a consistent macroscopic description is no longer given in the vicinity of non-equilibrium phase transitions (such as the deliquescence transition). Near the the phase transition (here the DRH), the turbulent fluctuations become as important as the mean values. Macroscopic values represent the most likely ones, which are identical with the mean values only if fluctuations can be neglected. In the real atmosphere, however, turbulent fluctuations are always present.

This result implies that the description of hygroscopic growth and shrinking during humidification and dehumidification of the ambient atmosphere, respectively, requires the consideration of both (i) the hysteresis effects during particle evolution and in addition (ii) the turbulent character of the thermodynamic conditions of the ambient atmosphere, which affect the macroscopic boundary conditions of phase transition. This is of particular importance, among others, for the purposes of atmospheric modeling as the optical properties of solid and deliquesced particles differ, which needs to be carefully considered, for example, in radiative transfer schemes in global atmospheric models (Haarig et al., 2017).

*Data availability.* Data sets will be made available via ACTRIS data centre in case of manuscript acceptance.

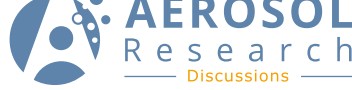

## Appendix A: Comparison of two different NaCl pre-conditioning settings and its effect on the RH field inside LACIS-T

Here, we compare the influence of two different types of NaCl particle pre-conditioning, i.e., case (i) and case (ii) as introduced in sect. 2.1, onto the RH field along the center line within the measurement section where the particles are injected, transported and later detected. The comparison is based on the LES performed in OpenFOAM. In both cases, the settings are $RH_A = 60\%$ and $RH_B = 85\%$ leading to a mean RH of 72.5% along the center line and about 78.000 monodisperse NaCl particles with a mass equivalent diameter of $Dp_{me} = 370$ nm are tracked inside the measurement section. The RH that the particles experience is shown in Fig. A1 in terms of median RH ($RH_{med}$), mean RH ($RH_{mean}$) and standard deviation of the RH ($\sigma_{RH}$). For both cases, $RH_{med}$ and $RH_{mean}$ fall together. The RH fluctuation distributions (not shown here) have a Gaussian shape. For case (i), the aerosol is injected with $RH \leq 5\%$ and it can be observed that $RH_{mean}$ in the mixing zone is lowered significantly close to the aerosol inlet due to the low aerosol RH. The obtained $RH_{mean}$ is about 46% which is 26.5% lower as it would occur in case of no particle injection. With increasing distance to the aerosol inlet, $RH_{mean}$ rises until about $z = 30$ cm distance where it starts to become constant. Due to this increase of $RH_{mean}$, the whole RH distribution gets shifted towards larger RH values which additionally influences $\sigma_{RH}$. This steep increase of $RH_{mean}$ combined with the increase of $\sigma_{RH}$ would complicate the data interpretation. Therefore, this setup was not used for our investigations.

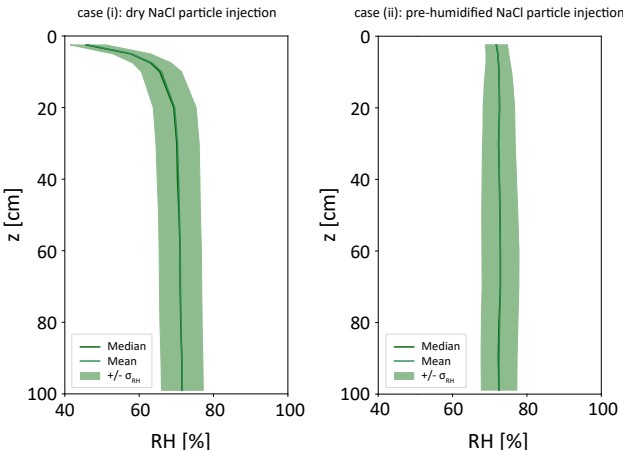

**Figure A1.** LES results for the scenario of $RH_A = 60\%$ and $RH_B = 85\%$ which results in a mean RH of 72.5% along the center region (i.e., the main mixing zone). The obtained median RH, mean RH and RH fluctuations in terms of $\pm\sigma_{RH}$ are plotted for the center region as a function of the distance $z$ to the aerosol inlet of the measurement section for two different types of particle injection. Left figure: Injection of dry aerosol (case (i): $RH \leq 5\%$ at $15°C$). Right figure: Injection of pre-humidified aerosol (case(ii): $RH = 70.2\%$ at $15°C$).





For case (ii), the aerosol is injected with RH $= 70.2\%$ which results in an only slightly lowered $\mathrm{RH_{mean}}$ in the mixing zone. $\sigma_{\mathrm{RH}}$ increases within the first 15 cm, becoming constant to a value of $4.9\%$. Therefore, this setup was applied for our experiments and data obtained for $z \geq 15$ cm was used for the later data interpretation.

**Appendix B: Comparison between measurements and simulations of $f_{\mathrm{del}}$ as a function of $t_{\mathrm{res}}$**

     In Figure B1, the measured and simulated deliquesced particle fractions $f_{\mathrm{del}}$ are plotted as a function of particle residence
time $t_{\mathrm{res}}$ exemplarily for $\mathrm{RH_{mean}} = 72.5\%$ and three different $\sigma_{\mathrm{RH}}$ values. In general, the time dependence of $f_{\mathrm{del}}$ can also be identified in the model results. However, the simulations tend to overestimate $f_{\mathrm{del}}$ for the cases of $\sigma_{\mathrm{RH}} = 4.9\%$ and $9.5\%$, and for residence times below 0.3 s. For the $\sigma_{\mathrm{RH}} = 2.8\%$ case, there is a closer agreement between the measurement and the simulation. However, the simulated slope differs from the experimentally determined one. This latter observation also holds for the other two cases.

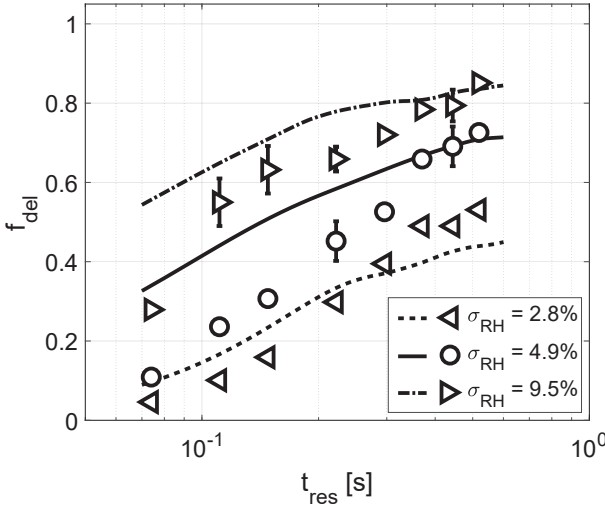

**Figure B1.** Measured (symbols) and simulated (lines) deliquesced particle fraction $f_{\mathrm{del}}$ as a function of particle residence time $t_{\mathrm{res}}$ for $\mathrm{RH_{mean}} = 72.5\%$.

The question arises about the reason for this discrepancy. In general, the LES model has proven to be an effective tool for simulating the thermodynamic and fluidic conditions inside the measurement section as well as the hygroscopic and dynamic growth of droplets which formed on NaCl particles (Niedermeier et al., 2020). In view of the lack of full information of the microscopic state of the investigated system, the modeling setup required some closure assumptions, which inheres uncertainties. Perhaps our assumption about the time scale of deliquescence ($\tau_{\mathrm{del}}$) is too simplified. We assumed - as mentioned before
- that NaCl deliquescence occurs on the time scale of $10^{-4}$ s once DRH $= 75.5\%$ is reached based on molecular dynamics





simulations performed by Bahadur and Russell (2008). An increase of the deliquescence time scale $\tau_{\mathrm{del}}$ to $10^{-3}$ s does not lead to a significant change of the simulated deliquesced particle fractions (not shown) because the mixing time scale $\tau_{\mathrm{mix}}$ (in the order of $10^{-1}$ s) is still two orders of magnitude larger than $\tau_{\mathrm{del}}$ so that the microphysical system is able to react on thermodynamic changes.

Several other theories of deliquescence have been developed during the last decades (e.g., McGraw and Lewis, 2009; Lamb and Verlinde, 2011; Hellmuth et al., 2013; Shchekin et al., 2013; Khvorostyanov and Curry, 2014) which are based on different approaches. For example, Khvorostyanov and Curry (2014) describe deliquescence as a nucleation process beginning with the formation of a liquid solution germ on a crystal surface. Consequently, they derive a nucleation rate in analogy to surface melting. From this nucleation rate, $\tau_{\mathrm{del}}$ for the deliquescence process of NaCl particles could be determined which depends,
among others, on the actual RH.

One could also think about an experimental approach in order to determine $\tau_{\mathrm{del}}$. However, for these measurements energy dissipation rates ranging from $10^{-2}$ up to $10^{6}$ m$^2$/s$^3$ would be needed so that the mixing time scale $\tau_{\mathrm{mix}}$ could be varied, ranging from values larger than $\tau_{\mathrm{del}}$ to values smaller than it. In the two extremes, the microphysical system will either be able react on thermodynamic changes (fast microphysics, $\tau_{\mathrm{mix}} \gg \tau_{\mathrm{del}}$) or not (slow microphysics, $\tau_{\mathrm{mix}} \ll \tau_{\mathrm{del}}$). The transition
region ($\tau_{\mathrm{mix}} \approx \tau_{\mathrm{del}}$) could give an estimate for the deliquescence time scale.

However, testing different theories and performing additional experiments for the derivation of $\tau_{\mathrm{del}}$ is beyond the scope of this study. It has to be the objective of future studies to investigate the suitability of different theories and experimental approaches in this context.

*Author contributions.* DeNi and RaHo wrote the manuscript with contributions from all co-authors. LACIS-T measurements and data eval-
uation were performed by DeNi and RaHo with contributions from SiSc, WiFr, MiPö and FrSt. Numerical simulations were performed by SiSc with contributions from DeNi, RaHo, FaSe, OlHe, and FrSt. All authors discussed the experimental and numerical results.

*Competing interests.* The authors declare that they have no conflict of interest.

*Acknowledgements.* We received funding for LACIS-T from ACTRIS-D which the German contribution to the Aerosol, Cloud, Trace Gases Research Infrastructure ACTRIS. ACTRIS-D is funded by the German Federal Ministry for Education and Research (BMBF) under grant
agreements 01LK2001A-K & 01LK2002A-G. Wiebke Frey has received funding from the European Union's Horizon 2020 RI programme as an Individual Fellowship under the Marie Skłodowska-Curie grant agreement number 835305.





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
