# Peer review of "Particle deliquescence in a turbulent humidity field"

_Aerosol Research, 2024_

## Author Comment (AC1)

Anonymous Referee #1, 02 Feb 2025

*This is an excellent paper. I have some items the authors may wish to consider, but there are no glaring errors or omissions.*

We thank referee 1 for the very positive feedback. We will address all comments in the following. The referee comments are in black, our response is given in blue color. Changes in the manuscript text will be highlighted.

*The authors cite books in several places. I'll use Seinfeld and Pandis as an example. That book is over 700 pages, and covers a wide variety of topics. Please, at least cite a chapter in the book. For example, in line 38 when you cite Seinfeld and Pandis for deliquescence, point to the paragraph or section in the book that is relevant for this.*

We followed the reviewer's suggestion and added book chapters and pages accordingly:

For the general citations about deliquescence, the following changes have been made:
lines 28, 35, 38-39: Khvorostyanov and Curry (2014) → Khvorostyanov and Curry (2014, Chapter 11, pp. 547-575),
lines 35, 38-39: Seinfeld and Pandis (2006) → Seinfeld and Pandis (2006, Chapter 10.2, pp. 449-461),
lines 38-39: Hellmuth et al. (2013) → Hellmuth et al. (2013, Chapter 12, pp. 317-347)

For the temperature dependence of the deliquescence RH, the following changes have been made:
lines 29, 53, 54, 224: Seinfeld and Pandis (2006) → Seinfeld and Pandis (2006, Chapter 10.2.2, pp. 453-454)
lines 29, 54: Khvorostyanov and Curry (2014) → Khvorostyanov and Curry (2014, Chapter 11.4.3, pp. 562-563)
For the theoretical description of deliquescence, the following changes have been made:
lines 284, 336, 337: Khvorostyanov and Curry (2014) → Khvorostyanov and Curry (2014, Chapter 11.3, pp. 553-558)
line 336: Hellmuth et al. (2013) → Hellmuth et al. (2013, Chapter 12.2, pp. 319-334)

Line 336: Lamb and Verlinde (2011) → Lamb and Verlinde (2011, Chapter 7.1, pp. 290-295)

*The section on particle generation, size selection, and pre-conditioning is comprehensive, but I think there needs to be some mention of doubly charged particles, because a DMA is used for size selection. Is the size used such that you don't have to worry about larger particles being present in the sample? Maybe having larger particles in the sample is not so important because you are considering deliquescence, not activation? A sentence explaining how the possibility of larger, doubly charged particles affect the results of the paper would be appreciated.*

Due to the use of the DMA, multiply charged particles can be present. There are two reasons why we chose particles with a mobility diameter of 400nm. First of all, these particles can be clearly detected by the welas 2300 sensor. Secondly, the amount of doubly charged particles is very low. Based on the particle size distribution resulting from the used atomizer, the amount of doubly charged particles is less than 1%. Actually, looking at Fig. 3 in the manuscript, it can be seen that there is a very small second mode for the deliquesced particles at about Dp = 1.3µm. Calculating the corresponding dry, solid NaCl particle size by means of the Köhler equation (RH_mean = 72.5%), yields a mass equivalent diameter of about 680 nm. This agrees well with the mass equivalent diameter of the doubly charged

particles, being about 650 nm. In conclusion, we argue that larger, doubly charged particles do not affect our results.

The following sentence has been added to the text: "For the experiments, we selected a mobility diameter of Dp_mob = 400 nm in order to be able to detect dry solid particles optically by means of a Promo 2000 with welas 2300 aerosol spectrometer (Palas GmbH, Karlsruhe, Germany) inside LACIS-T, **as well as to minimize the amount of larger, doubly charged particles being present. As it turns out, their proportion to the total number of selected particles is less than 1%, i.e., doubly charged particles do not affect our results.**"

*Line 187: I know this is picky… "About 78.000…"  That's five significant figures. I recommend "78".*

We made a mistake here. It should read seventy-eight thousand particles. So, the dot as a separator is wrong. It was changed accordingly to 78000.

*I was particularly interested in the discussion of the time for deliquescence in the appendix. I think deliquescence is a nucleated phase transition, as noted by the authors when they cite Khvorostyanov and Curry. See also Lu et al (2008) and Cantrell et al (2002). I am not convinced that the authors are seeing evidence for nucleation in their experiments though. I would expect salt particles resulting from efflorescence of an atomized solution to be defect rich, which would lower the nucleation barrier to a value that I doubt you would detect it in these experiments.*

The referee raised a valuable point. We agree that we are not able to observe/resolve the nucleation process itself in our lab experiments. However, this is also not the focus of our study. For us, it is important whether the NaCl particles are solid or deliquesced, and that we are able to distinguish between both phase states.

The model study was performed to test whether or not we are sensitive enough to observe the effects of turbulent RH fluctuations on the number of deliquesced particles. With the – probably too simplified – assumption of a constant time scale of deliquescence, we found a qualitative agreement between lab observations and model results. A Quantitative agreement could not be achieved. A possible explanation for this missing agreement could be the too simple treatment of deliquescence in the model. However, testing the applicability of different deliquescence models is not in the focus of our paper, but could be the aim of future studies. This is already mentioned in the original manuscript. However, we added a sentence at the beginning of Appendix B:
**"The model study was performed to test whether or not we are sensitive enough to observe the effects of turbulent RH fluctuations on the fraction of deliquesced particles.";**
as well as a sentence at the end of Appendix B: **"However, it should be noted that we were not able to observe / resolve the nucleation process itself in our laboratory experiments."**

References:

Cantrell, W., McCrory, C. and Ewing, G.E., 2002. Nucleated deliquescence of salt. The Journal of chemical physics, 116(5), pp.2116-2120.

Lu, P.D., He, T. and Zhang, Y.H., 2008. Relative humidity anneal effect on hygroscopicity of aerosol particles studied by rapid-scan FTIR-ATR spectroscopy. Geophysical research letters, 35(20).

---

## Author Comment (AC2)

Anonymous Referee #2, 11 Feb 2025

*This manuscript is well-prepared and presents the cloud simulator experiment combined with Large-eddy simulation to investigates the deliquesce behavior of NaCl particles under turbulent humidity conditions. The result highlights the importance of mean RH, the strength of RH fluctuations, and the residence time of particles on the deliquescence process. While these results are generally expected, previous studies have not provided clear experimental evidence, making this study a valuable contribution that fills a critical knowledge gap. The manuscript is well organized, and the data is presented clearly. I recommend acceptance after considering my comments below, which should further improve the quality of this manuscript.*

We thank referee 2 for the positive feedback. We will address all comments in the following. The referee comments are in black, our response is given in blue color. Changes in the manuscript text will be highlighted.

*Comments:*

*Although the selection of Dp 400 nm particles may minimize the impact of multiply charged particles, it depends on the number size distribution of generated particles from the atomizer. A brief explanation is needed.*

Indeed, the amount of doubly charged particles is very low. As mentioned by the referee, the amount depends on the number size distribution of the generated particles generated by the atomizer. For our setting, the amount of doubly charged particles is less than 1%. This is also confirmed by the deliquescence measurements itself. Looking at Fig. 3 in the manuscript, it can be seen that there is a very small second mode in the distribution of the deliquesced particles at about Dp = 1.3µm, which results from the doubly charged particles. However, with this mode a) representing only approximately 1% of the particle population, and b) being excluded from the analysis, we conclude that larger, doubly charged particles do not affect our results.
The following sentence has been added to the text: "For the experiments, we selected a mobility diameter of Dp_mob = 400 nm in order to be able to detect dry solid particles optically by means of a Promo 2000 with welas 2300 aerosol spectrometer (Palas GmbH, Karlsruhe, Germany) inside LACIS-T, **as well as to minimize the amount of larger, doubly charged particles being present. As it turns out, their proportion to the total number of selected particles is less than 1%, i.e., doubly charged particles do not affect our results.**"

*In the dehumidified scheme (Case 3), the particle shape may need further correction. This could also slightly affect the size distribution in Figure 3. Please refer to Biskos et al. (2006). If it is negligible, please clarify it.*

We are not completely sure what the referee meant here. The particles in case 3 are not dehumidified, i.e., they are actually humidified to RH = 100% at 12.5°C so that all particles deliquesce before entering LACIS-T. In that case, the particles should become spherical. It can be seen in Fig. 3 in the original manuscript, that the size distribution of the humidified particles (graph (3)) is much narrower compared to the solid NaCl particles (graph (1)).
However, in case 1, the particles are dried to RH < 5%.  We assumed a shape factor of 1.08 (Kelly and McMurry, 1992) for the conversion from mobility to the mass equivalent diameter. Our measurements, presented in Fig. 3, graph (3), show that the calculated wet diameter at RH_mean =

72.5% agrees well with the measured wet diameter, suggesting that the shape factor assumed for the dry NaCL properly accounts for the non-sphericity of the dry NaCL particles within the measurement uncertainty. This is already mentioned in the text.

Our dry mobility diameters of 400 nm are also much above the upper limit of the size range of the 6-60 nm dry mobility diameters of NaCl particles studied by Biskos et al. (2006), who found the growth factors steadily decreasing for dry sizes below 40 nm. While we agree with this finding (inclusive the statements on its causing physical factors) we do not see indications for the need of a corresponding size correction to the graphs in Fig. 3, especially graph (3).

Finally, and most importantly, we can also state that the shape of the optically detected size distributions is of secondary importance for the determination of the deliquesced particles fractions – which is the focus of our study – as long as the two modes, solid and deliquesced particles, can be separated clearly. This is always the case in our study. From the above considerations we conclude, that a size correction as suggested by the referee is not required.

Nothing has been changed in the manuscript.

*This study fully relies on simulated RH fluctuations. While this is reasonable given instrumental limitations, I would like to see a discussion on the accuracy of these simulated fluctuations and the possible uncertainties associated with them.*

We are performing large eddy simulations (LES), i.e., the most relevant scales of the turbulent flow field are resolved and only the smallest, homogeneous scales have to be parameterized/modelled. Looking on Fig. 5 in Niedermeier et al. (2020), it can be seen that the turbulent flow characteristics inside LACIS-T can be appropriately simulated via LES. The same holds true for heat and mass transfer, as shown in Figs. 6-9 in Niedermeier et al. (2020).

Concerning possible uncertainties, those would presumably origin from the non-resolved sub-grid scales (SGS). To minimize these uncertainties, we apply a very high resolution, especially underneath the cutting edge with grid lengths of 1-3 mm, in order to capture even the small-scale processes. For comparison, the Kolgomorov length is between $0.5 - 1.4$ mm in our case. Based on our ongoing work on the influence of the SGS on the fluctuations and in accordance to a study performed by Chandrakar et al. (2022), we estimate the contributions from the SGS fluctuations to the total fluctuations to be max. +/- 0.04 x $\sigma_{RH}$. The following text has been added to sect. 4.1:

**"Note that there is an influence of the SGS motions onto the simulated fluctuations and consequently sigma_RH. Based on currently ongoing work and in accordance to a study performed by Chandrakar et al. (2022), we estimate the contributions from these SGS fluctuations to the total fluctuations to be max. +/- 0.04 x $\sigma_{RH}$, which represents the uncertainty of the determined $\sigma_{RH}$ values."**

*The deliquescence time scale is assumed to be $10^{-4}$ s in simulation. However, in real ambient conditions, for example, for organic/inorganic mixed particles, it can reach equilibrium on the order of seconds (Duplissy et al., 2009). While inorganic particles typically deliquesce in less than 1 s, I suspect that a value of $10^{-4}$ s may be too short for realistic atmospheric conditions. Could the authors explore how different τdel values affect their results?*

In the original manuscript, the topic of different $\tau_{del}$ values has already been dealt with in Appendix B. In there it was stated: "*An increase of the deliquescence time scale $\tau_{del}$ to $10^{-3}$ s does not lead to a significant change of the simulated deliquesced particle fractions (not shown) because the mixing*

*time scale τmix (in the order of 10^−1 s) is still two orders of magnitude larger than τ_del so that the microphysical system is able to react on thermodynamic changes."*

A further increase of τ_del, so that it is in the range of, or larger than τ_mix, would affect the results, presumably leading to a decrease in the deliquesced particle fractions in the simulations. However, the focus of this study is to demonstrate experimentally that the fraction of deliquesced particles depends on mean RH, RH fluctuations and residence time. With the – probably too simplified – assumption of a constant time scale of deliquescence, we found a qualitative agreement between lab observations and model results. Quantitative agreement could not be achieved. A possible explanation for the missing quantitative agreement could be indeed the too simple treatment of deliquescence in the model. However, testing the applicability of different deliquescence models is not in the focus of our paper, but would be the aim of future studies. This is already mentioned in the original manuscript.

*How do the RH fluctuations intensities in this experiment compare to real atmospheric environments? Some discussion about it could improve the scope of this study.*

A few studies exist which report turbulent RH fluctuations. Kulmala et al. (1997) derived RH standard deviations from two field studies (Lenchow et al., 1994 – aircraft measurements over a warm ocean surface in winter; MacPherson et al., 1992 – atmospheric boundary layer aircraft measurements for overcast and "few cumuli" conditions). In summary, they obtained standard deviations in the range of 1% to 4.6%, depending on the environmental conditions. Field measurements in the Netherlands close to Utrecht performed in May 2008 (Siebert and Shaw, 2017) found standard deviations of the RH distribution to be ca. 2.4% at ground level, ca. 2.3% in regions outside of clouds (ca. 1100m above ground) and about 1.5% inside a developing cumulus cloud (i.e., being in the range reported by Kulmala et al. (1997)). Concerning the latter value, it turns out that the growth or evaporation of cloud droplets reduce the magnitude of the RH fluctuations (Siebert and Shaw, 2017).

In summary, the available field measurements show sigma_RH values which were covered in our investigation, indicating the importance of the results of our study, as we can observe a distinct influence of the turbulent RH fluctuation on the fraction of deliquesced particles at these sigma_RH values. Furthermore, when looking on the number of available field observations, we suggest to collect additional data on atmospheric RH fluctuations, to be carried out in terrestrial and marine environments, at ground level and above.

Corresponding paragraphs are added to the results section as well as the summary and conclusion section:

**"A few studies exist (e.g., MacPherson et al., 1992; Lenchow et al., 1994; Kulmala et al., 1997; Siebert and Shaw, 2017), which show σ_RH values in the range of 1% to 4.6%, depending on the environmental conditions. Our investigations cover this range of observed RH fluctuations and we observe a distinct influence of the turbulent RH fluctuation on the fraction of deliquesced particles at these σ_RH values. This indicates that our results are relevant for the atmosphere."**
and
**"In that sense, we also suggest to collect additional data on atmospheric RH fluctuations, to be carried out in terrestrial and marine environments, at ground level and above."**

*Minor issue:*
*Line 41: Do you mean that "most" experiments were performed under laminar flow conditions?*

Looking into the literature, there are many different methods for the investigations of deliquescence, including e.g., particles on substrates, filter-based analyzers, optical, electron and X-ray microscopy, electrodynamic balances, levitation, H-TDMAs, laminar flow tubes" (e.g., Tang et al., 2019). So, the number of methods/techniques applying a flow in general is low compared to all other methods/techniques and if a flow is involved, it has been laminar (so far, no measurements under turbulent conditions).
We changed the corresponding sentence to: **"**Most of the experimental investigations have focused on the process itself using various techniques (Tang et al., 2019). **A majority of experiments were carried out under no-flow conditions. Continuous flow type experiments (such as in Wex et al., 2007) were carried out under laminar flow conditions."**

References

Biskos, G., et al. "Nanosize effect on the hygroscopic growth factor of aerosol particles." *Geophysical Research Letters* 33.7 (2006).

Chandrakar, K. K., Morrison, H., Grabowski, W. W., Bryan, G. H., and Shaw, R. A. (2022) Supersaturation variability from scalar mixing: Evaluation of a new subgrid-scale model using direct numerical simulations of turbulent Rayleigh–Bénard convection, *Journal of the Atmospheric Sciences*, 79(4), 1191–1210, https://doi.org/10.1175/JAS-D-21-0250.1.

Duplissy, Jonathan, et al. "Intercomparison study of six HTDMAs: results and recommendations." *Atmospheric Measurement Techniques* 2.2 (2009): 363-378.

Kulmala, M., Rannik, Ü., Zapadinsky, E. L., and Clement, C. F. (1997). The effect of saturation fluctuations on droplet growth. *Journal of Aerosol Science*, *28*(8), 1395-1409, https://doi.org/10.1016/S0021-8502(97)00015-3.

Lenchow, D. H., Mann, J. and Kristensen. L. (1994). How long is long enough when measuring fluxes and other turbulence statistics'?. *J. Atmos. Oceanic Technol.*, 11, 661-673.

MacPherson, J. I., Grossman, R. L. and Kelly, R. D. (1992) Intercomparison results for FIFE flux aircraft. *Journal of Geophysical Research: Atmospheres*. 97, (D17), 18,499-18,514, https://doi.org/10.1029/92JD00272.

Niedermeier, D., Voigtländer, J., Schmalfuß, S., Busch, D., Schumacher, J., Shaw, R. A., and Stratmann, F. (2020) Characterization and first results from LACIS-T: a moist-air wind tunnel to study aerosol–cloud–turbulence interactions, *Atmospheric Measurement Techniques*, 13, 2015–2033, https://doi.org/10.5194/amt-13-2015-2020.

Siebert, H., and Shaw, R. A. (2017). Supersaturation fluctuations during the early stage of cumulus formation. *Journal of the Atmospheric Sciences*. 74(4), 975-988, https://doi.org/10.1175/JAS-D-16-0115.1.

Tang, M., Chan, C. K., Li, Y. J., Su, H., Ma, Q.,Wu, Z., Zhang, G.,Wang, Z., Ge, M., Hu, M., He, H., and Wang, X. (2019), A review of experimental techniques for aerosol hygroscopicity studies. *Atmospheric Chemistry and Physics*, 19, 12 631–12 686, https://doi.org/10.5194/acp–19–12 631–2019.

Wex, H., Ziese, M., Kiselev, A., Henning, S., and Stratmann, F. (2007), Deliquescence and hygroscopic growth of succinic acid particles measured with LACIS. *Geophysical Research Letters*, 34, L17810, https://doi.org/10.1029/2007GL030185.